# Popular Dietary Trends’ Impact on Athletic Performance: A Critical Analysis Review

**DOI:** 10.3390/nu15163511

**Published:** 2023-08-09

**Authors:** Matthew Kaufman, Chantal Nguyen, Maya Shetty, Marily Oppezzo, Michelle Barrack, Michael Fredericson

**Affiliations:** 1Department of Orthopaedic Surgery, Stanford University, Redwood City, CA 94063, USA; 2Prevention Research Center, Stanford University, Redwood City, CA 94063, USA; 3Department of Family and Consumer Sciences, California State University, Long Beach, CA 90840, USA

**Keywords:** athletes, nutrition, diets, vegan, plant-based, ketogenic, Mediterranean diet, intermittent fasting, eating disorders

## Abstract

Background: Nutrition fuels optimal performance for athletes. With increased research developments, numerous diets available, and publicity from professional athletes, a review of dietary patterns impact on athletic performance is warranted. Results: The Mediterranean diet is a low inflammatory diet linked to improved power and muscle endurance and body composition. Ketogenic diets are restrictive of carbohydrates and proteins. Though both show no decrements in weight loss, ketogenic diets, which is a more restrictive form of low-carbohydrate diets, can be more difficult to follow. High-protein and protein-paced versions of low-carbohydrate diets have also shown to benefit athletic performance. Plant-based diets have many variations. Vegans are at risk of micronutrient deficiencies and decreased leucine content, and therefore, decreased muscle protein synthesis. However, the literature has not shown decreases in performance compared to omnivores. Intermittent fasting has many different versions, which may not suit those with comorbidities or specific needs as well as lead to decreases in sprint speed and worsening time to exhaustion. Conclusions: This paper critically evaluates the research on diets in relation to athletic performance and details some of the potential risks that should be monitored. No one diet is universally recommend for athletes; however, this article provides the information for athletes to analyze, in conjunction with medical professional counsel, their own diet and consider sustainable changes that can help achieve performance and body habitus goals.

## 1. Introduction

Nutrition and athletes are inextricably linked. Nutrition fuels the body for optimal performance and promotes recovery efforts, as supported by the evidence-based positions of the American Dietetic Association, Dietitians of Canada, and the American College of Sports Medicine [1,2]. These organizations recommend selections of food and fluid that provide athletes with adequate energy and nourishment to sustain high-intensity training, promote recovery, and support a body habitus conducive to their sport.

Athletes of all levels, youth to professional, are turning to nutrition to improve the trajectory of their performance. These competitors encounter a wide range of nutritional recommendations and diet trends through exposure from social media, peers, and coaches, among other sources [3]. 

The role of this review is to critically evaluate the current literature addressing these diets, describe the content of the diet, feasibility for adherence, potential pitfalls, and impact on performance in sport. Several of the most frequently addressed diets studied among athletes include the Mediterranean diet, low-carbohydrate and ketogenic diets, vegetarian and plant-based diets, as well as intermittent fasting. In addition, when considering dietary lifestyle changes that athletes can adopt, it is crucial to consider the possible maladaptive changes from unhealthy versions of these diets as well as eating disorders.

## 2. Materials and Methods

The current study is a narrative review to address diets with the most robust research evidence in humans regarding dietary impacts on athletic performance. While this article does not comprehensively discuss all diets and their associations to athletic performance, it critically evaluates the current level of evidence for the five most common diet patterns described in the literature from the PubMed search engine. Additionally, given the concerns to develop eating disorders among athletes following a restrictive diet, additional articles were reviewed to describe a proposed initial risk assessment for the development of eating disorders among athletes. The authors evaluated narrative and systematic reviews as well as meta-analyses of these common dietary practices from 2015–2022 on PubMed, and subsequently evaluated the primary literature that was cited within these works to develop the evidence referenced in the current study.

## 3. Results and Discussion

### 3.1. General Recommendations for Diets for Athletes

Athletes’ dietary intake should support the demands of exercise, recovery, training adaptations, and overall health and should, thus, provide a sufficient level of calories, carbohydrate, protein, dietary fats, fiber, fluids, and essential vitamins and minerals [1,2]. Diets that increase risk of energy deficiency or contribute to below recommended intakes of carbohydrates, protein, fluids, or other essential nutrients may deter athletes’ health or performance and should be avoided [1,2]. Therefore, an overview of the diets will be provided and evaluated based on prior evidence for either optimizing or limiting energy or nutrient intake in athletes and subsequent potential outcomes. 

Athletes in a caloric deficit or those not engaging in strength training are at risk for losing muscle mass or limited muscle gains during resistance training; both are critical to performance [4,5]. Caloric deficits also lead to increased risks of bone mineral disease and overall decreased bone health in athletes [6]. Protein needs for individuals engaging in moderate-to-high-intensity exercise exceed the RDA of 0.8 g/kg/day and typically range from 1.2–2.0 g/kg/day [1,7]. Meeting these needs by consuming protein-rich foods at regular intervals (every 3–4 h) throughout the day and post-exercise may optimally support protein synthesis, recovery, and performance outcomes [1,7]. Carbohydrates serve as the preferred energy source for moderate to high-intensity exercise with increased needs (up to 10–12 g/kg/day) for endurance or ultra-endurance athletes participating in high-volume training [1]. 

Essential vitamins and minerals support exercise training and performance by facilitating energy metabolism (i.e., B-vitamins, magnesium), promoting bone health (calcium, vitamin D), erythropoiesis, and oxygen delivery (i.e., iron, folate, B12), or serving an anti-inflammatory role (i.e., antioxidants, EPA/DHA) [1,8]. Adequate hydration is key to sustaining numerous processes affecting athletic performance [1]. Sweat loss with exercise disrupts the fluid balance, and fluid losses ≥2% can alter performance. Thirst may not adequately reflect fluid needs with exercise and needs largely depend on sweat rate. Given this, while general daily fluid needed for females and males range from 2.7–3.7 L/day, for athletes, individualized assessments and recommendations are needed [1,9]. 

Some popular diets may limit one or more key nutrients for athletes. Given the relationship between glycogen levels and performance outcomes, evidence does not support a benefit of following a low-carbohydrate diet for most athletes [1]. Several carbohydrate-rich food sources (i.e., whole grains, legumes, fruit, vegetables, milk/yogurt) contain essential nutrients that support energy metabolism and bone health, and serve an antioxidant role. Intermittent fasting restricts the hours of eating during the day, which can disrupt nutrient timing goals that emphasize regular intervals of food intake during waking hours [10]. Intermittent fasting can reduce an athlete’s overall intake of calories due to skipping meals, which may lead to an energy deficit [10]. 

Diets restricting animal products (i.e., vegan/vegetarian) may be limiting in some essential amino acids [11] and essential micronutrients that support athlete health and performance, such as iron, zinc, calcium, and vitamin D. While these micronutrients (i.e., calcium and iron) can be consumed through plant-based foods, nutrients may be less bioavailable. This could be a major consideration for those transitioning to a diet avoiding animal products. Leucine can be provided by soy, corn, pea, brown rice or potato, though higher quantities of these foods are needed to provide an equivalent leucine content [11]. Other considerations include the level of ultra-processed foods (UFPs), which may be higher among individuals following a vegetarian or vegan diet, and linked to adverse health outcomes [12,13]. Furthermore, ketogenic diets contain higher levels of red meat and saturated fat and increased LDL cholesterol [14].

Changes to the diet can be difficult for athletes. Gradual, impactful changes with behavioral supports such as gradual introduction or subtraction of foods or calories can lead to longer sustained compliance and turn diet changes into lifestyle adaptations, similar to those trying to lose weight. A recent meta-analysis showed that continued gradual change to lifestyle lead to slowed weight regain, a common outcome from weight loss diet strategies [15]. When making these gradual changes, it is crucial to understand what choices are available to athletes while considering each choice’s risks and benefit.

### 3.2. Mediterranean Diet 

The Mediterranean diet (MedDiet) consists of lean proteins, such as fish and chicken, vegetables, berries, olive oil, and nuts, and has been practiced in Mediterranean countries for centuries [16]. It was characterized in the 1960s by the North American biochemist and nutritionist Ancel Keys, who lived and conducted research in the Cilento Coast of Southern Italy for over 40 years [17]. It was originally touted in the Seven Countries Study to reduce the risk of cardiovascular disease compared to the United States and Northern European countries over a 25-year span [18]. While the diet has developed numerous variations over the years, the basic tenets are the same. The Mediterranean Diet Foundation (2011) and 1999 Greek Dietary Guidelines recommend at least 2 servings of vegetables every meal, 1–2 servings of fruit per meal, 1–2 servings of wholegrain bread per meal, at least 2 weekly servings of legumes, 3–4 weekly servings of nuts, at least 2 weekly servings of fish or seafood, 2–4 weekly servings of eggs, 2–4 weekly servings of poultry, 2 servings daily of dairy, and less than 2 servings per week of sweets and red meat, with the main added lipid being olive oil and red wine in moderation [19]. The recommended serving sizes are 25 g of bread (~1 slice of store bought bread), 100 g potatoes (a small potato), 50–60 g of cooked pasta (1/4 cup of cooked pasta), 100 g of vegetables (a palm size of broccoli), 80 g of apple (half an apple), 60 g of banana (half of a medium sized banana), 100 g of orange (~1 orange), 200 g of melon (~1 cup), 30 g of grapes (~10), 1 cup of milk or yogurt, 1 egg, 60 g of meat (chicken, beef, or seafood) (~palm of your hand), and 100 g of cooked dry beans (~half cup) [19]. The MedDiet has been shown to help with numerous chronic diseases, cancer, and cognitive performance, and is now being studied in trained athletes due to its low inflammatory index as well as its ability to support sustained training and recovery. Studies show that as adherence to the MedDiet increases inflammatory markers, such as C-reactive protein, tumor necrosis factor-A, and plasminogen activator inhibitor-1, and decreases rates of metabolic syndrome when compared to the Western diet, which increases inflammatory markers [20,21].

For performance, the MedDiet has been touted to help athletes from a performance and body composition standpoint, as it provides nutritional support for their training and athletic demands with a low inflammatory index. During an 8-week MedDiet intervention, CrossFit athletes had significant increases in squat jump performance, power, muscular endurance, and anaerobic power [22]. Additionally, athletes saw increases in CrossFit-specific maneuvers and performance [22]. The athletes in the dietary group showed increased performance during the chin-up test to exhaustion, i.e., from 11 repetitions to 13 repetitions [22]. Similarly, after 3 months of adhering to a MedDiet, University of Sports Center Bergamo kickboxers had statistically significant increases in squat strength and the countermovement jump test (CMJ), a barometer for lower body power (the CMJ test is a validated test that correlates to 1 RM maximal strength and sprinting ability [23]). In the same study, runners’ VO2max increased [24]. Of note, neither of these studies had a control group, but a strength of the within-subject design is that athletes saw changes relative to their own performance after making changes to their diets.

Body composition serves as one of several factors that influences athlete’s training and performance. The MedDiet may support the goals of the athlete in a moderate caloric deficit, while still providing adequate nutrition to support exercise training and performance gains while promoting weight loss if necessary. Adequate nutrition is critical to a sport such as female gymnastics, which includes long hours of training, starting at a young age, with high metabolic demands for growth. A study investigating adolescent gymnasts age 11–18 showed that those with a higher KIDMED score, a pediatric-adapted Mediterranean Diet Quality Index that measures adherence to the MedDiet, had a healthier body mass index compared to those that had lower adherence [25]. While on the MedDiet, it is crucial that all athletes, especially those at risk for malnutrition, especially gymnasts, runners, and dancers, where pressures exist to maintain a low body weight, consume the calories that they need to maintain a healthy weight. This study highlights that athletes can benefit from nutritional intervention, as it has been well documented that athletes oftentimes do not have adequate nutrition to supply their metabolic demands and benefit from nutritional guidance and intervention [26,27]. Secondly, the MedDiet likely can help athletes improve performance and body composition because it provides both the nutritional support that high-performing athletes need to sustain performance while providing a low-inflammatory diet rich in antioxidants and essential vitamins and minerals. 

While adherence to diet in research and in practical application is difficult to evaluate, evidence supports benefits of high adherence to the MedDiet. Many different research tools have been developed for assessment for adherence, including Medi-Lite, MDSS, and MEDAS, as well as the KIDMED for the pediatric/adolescent population [28,29,30,31]. Through these tools, it has been demonstrated that athletes have higher adherence then the general population [32].

There were initial concerns that increased dietary discipline would increase the risk of sport burnout; however, a recent study shows that there was no statistically increased risk of sport burnout among adolescent tennis players with high adherence to the MedDiet [33]. Further studies are needed to assess risk of burnout and ability to adhere to the MedDiet among athletes undergoing extensive training among all levels of sport. 

The MedDiet covers the nutritional needs of most athletes due to its well-balanced composition of lean proteins, complex carbohydrates, and unsaturated fats. Overall, the strengths of this diet are that it is high in foods that can support the high energy demands of athletes combined with a low inflammatory index that promotes recovery [34,35]. The low inflammatory index of the MedDiet stems from its high omega-3 fatty acid content, high antioxidant (flavonoids, polyphenols, and phytosterols) content, high amount of fruits and vegetables, and low amount of red meat [36]. As discussed, athletes following the MedDiet have seen increases in aerobic and anaerobic performance, suggesting this diet provides benefits to power and endurance athletes and additionally does not contribute to burnout, with researched adherence tools. Additionally, the MedDiet is associated with improvements in body composition, which, along with adequate nutrition, are especially important for peak performance capacity during long training or competitions. Finally, the wide range of foods within the MedDiet allow for higher adherence compared to most diets, making it an optimal choice for athletes who want increased nutritional options.

### 3.3. Ketogenic and Low-Carbohydrate Diets

A ketogenic diet is very low in carbohydrate, moderate in protein, and high in fat, with the classical ketogenic diet composed of a 3–4:1 ratio of dietary fat to carbohydrate and protein [37]. Several variations of the ketogenic diet contain up to 10% carbohydrate, 75–90% fat and 10% protein. For some versions, this could equate to as little as 30–50 g of carbohydrates per day [38]. While less restrictive, these versions still provide significantly lower levels of carbohydrate as compared to sports-specific recommendations. A ketogenic diet aims to promote a state of ketosis, a fasting state with low carbohydrate availability promoting the formation of ketone bodies, including acetoacetate, acetone, and beta-hydroxybutyrate [39].

The ketogenic diet was born out of an idea by Bernarr Macfadden, a physical fitness guru/cultist in the early 20th century, that claimed fasting for 3 days to 3 weeks could cure any disease [40]. This idea of fasting showed efficacy in decreasing the amount of seizures in those with epilepsy as early as the 1920s [40]. The reason that this works is because ketones are metabolized much more slowly than glucose and are not the brain’s and red blood cells’ preferred food sources; as a result, this leads to a relatively starved state of brain tissue, leading the brain to conserve energy and be less likely hyperactive, as is the case in a seizure [41]. The only way that this is truly efficacious is if the primary macronutrient in the diet is fat, as even protein needs to be limited to limit the potential for gluconeogenesis. In 1921, Dr. Wilder of the Mayo Clinic proposed a restrictive diet that acted as an alternative to complete fasting that decreased epileptic seizures: the ketogenic diet [40]. In 1925, the Mayo Clinic stated that the ketogenic diet consisted of 1 g of protein per kg of body weight, 10–15 g of carbohydrates per day, and the remainder of calories coming from fat, similar to the framework of the diet today [40]. This diet became increasingly popular in the field of athletics when Phinney et al. published his study in 1983, stating that well-trained cyclists could sustain exercise capacity at submaximal capacity for 21–28 days while on the ketogenic diet [42]. This work, as well as subsequent papers, suggested that the ketogenic diet could slow the oxidation of carbohydrate stores, thereby keeping glycogen stores higher for longer maintenance of maximum or near-maximum performance [43]. 

A randomized crossover study, with participants having carbohydrates ad libitum diet and carb restricted <10% daily each for 3 months, suggested that a ketogenic diet (<50 g or <10% carbohydrate intake/day, 20% protein) over 12 weeks could be efficacious for sports involving weight-class, specifically powerlifting and Olympic weightlifting, by reducing body mass and preserving baseline performance in resistance athletes. There was a statistically significant decrease in body mass and lean mass reported for athletes on a KD versus an ad libitum diet (>250 daily grams of carbohydrates)—presumably in the setting of decreased glycogen storage and increased activation energy to catabolize protein—all without negatively impacting lifting performance, blood glucose, and electrolyte balance [44]. Further studies defend this principle that a KD does not cause further athletic decrement despite decreased body mass. In eight trained endurance athletes on a four-week KD (4% carbohydrate, 78% fat, 18% protein), there was no difference in time to exhaustion (TTE) in vigorous exercise (70% VO2 max) when compared to athletes consuming a higher carbohydrate diet (43% carbohydrate, 38% fat, 19% protein) [45]. When increased to near-maximal to maximal intensity (>85% of one-repetition maximum), there was no decrement in back squat, bench press, clean, jerk, and deadlift performance as observed in a randomized control trial following trained athletes adhering to KD (5% carbohydrate, 75% fat, 20% protein) for 10–12 weeks [46]. 

When further assessing the role of aerobic capacity with a KD, another meta-analysis including 10 studies found no significant difference in performance or maximum aerobic capacity, namely VO2 max, time to exhaustion, maximum heart rate, or rating of perceived exertion (RPE) during treadmill tests with athletes adhering to KD, though a significant reduction in respiratory exchange ratio (RER) was observed [46]. RER is directly correlated with carbohydrate oxidation—the higher the RER, the more carbohydrates are being oxidized as opposed to fat, which theoretically decreases overall oxygen efficiency in athletes. Though this could play a small role in performance, ongoing literature suggests KD has minimal impact on maximum aerobic capacity. Interestingly though, KD has been shown to prolong professional athlete careers by enhancing weight loss [47] and may have variable outcomes based on gender. In a study of 22 CrossFit athletes (11 male, 11 female) over a four-week period, a KD (>75% of daily energy from fats, up to 5% from carbohydrates) led to increased fat oxidation in both genders and decreased oxidation of carbohydrates in male CrossFit athletes. Specifically in males, fat oxidation was statistically significant at lower VO2 maxes than fat oxidation in female CrossFit athletes. Compliance to the diet had a non-significant difference, but it was noted that eight subjects (1/3 of the participants) did not complete the study (two due to injury, six due to non-adherence to diet or failure to achieve ketosis) [48].

It is important to note that athletes undergoing KD while concomitantly not meeting daily caloric needs may have decreased body mass in the form of decreased muscle mass, thereby leading to some performance decrements in strength training. In addition, KD may also negatively impact bone health in endurance athletes. In a study assessing serologic markers of bone health homeostasis in male and female world-class race walkers on a KD for 3.5 weeks, carbohydrate restriction led to increased serologic markers, at rest and after exercise, of bone breakdown (cross-linked C-terminal telopeptide of type I collagen), formation (procollagen 1 N-terminal propeptide), and metabolism (osteocalcin) [49]. Similar results were seen in a separate parallel group design of 28 elite racewalkers—those allocated to the low-carbohydrate group had significantly higher serologic markers of bone breakdown, which improved with increasing carbohydrate intake [50]. This poses significant considerations for athletes deciding on a particular diet for performance, as KD can have a compounded effect on damaging bone health—which can already be at high risk in athletes with the male or female athlete triad. 

As another consideration, there are possible negative physiologic sequelae from KD, particularly in iron, immunologic, and stress responses to exercise in endurance athletes. A study of elite male race walkers who completed a 25 km race walk highlighted this concern—the athletes first engaged in Phase 1 of the study, where they consumed a high-carbohydrate diet, high-energy diet (65% CHO, 15% protein, and 20% fat, target energy availability 40 kcal·kg^−1^ FFM·day^−1^); then, athletes were randomized to Phase 2, in which they either continued the high-carbohydrate diet, transitioned to a low-carbohydrate diet (<50 g CHO per day), or transitioned to a low-energy-availability diet (60% of energy from CHO, 25% of energy from protein, 15% of energy from fat, target energy availability 15 kcal·kg^−1^ FFM·day^−1^). Those who transitioned to the low-carbohydrate diet had higher levels of interleukin 6, white blood cell count, hepcidin, and cortisol, compared to baseline. These serologic markers serve as surrogates for the deleterious effects of KD on physiologic concentrations of iron metabolism (hepcidin), stress (interleukin 6, cortisol), and inflammation (white blood cell count, interleukin 6) in the body [51].

With these decrements to health noted, it is generally recommended to avoid chronic carbohydrate restriction during intense training periods, especially for endurance athletes [52,53]. The subcellular physiologic changes that occur with severe carbohydrate restriction manifest not only through deleterious effects in muscle strength, but also increase the general oxygen cost of exercise, which can negatively impact athletic performance in the acute phase as well as longitudinally [54,55]. Thus, there is generally lower overall adherence to exercise, decreased exercise tolerance, higher baseline heart rates, and a higher rate of perceived exertion in athletes on a KD given the elevated blood ketone levels and downstream biochemical effects of increasing the stress response within the body—the summation of these effects make it important to carefully consider the risks and benefits of choosing this diet [56]. However, another significant factor in choosing this diet is possible complications with compliance. Generally, it has been observed that KD compliance is difficult given the strict carbohydrate percentage cutoffs, often leading participants to increase daily caloric intake or deviate from the scheduled diet [57]. Landry et al. showed that adherence to KD was statistically significantly lower than that of the MedDiet and a low-fat diet that was used in his study of diabetics [58]. While this is a markedly different population than the trained athlete, it demonstrates the difficulty of adhering to a diet with fat as a primary fuel source [58]. 

Thus, athletes’ adherence to KD theoretically optimizes fatty oxidation as primary fuel for activity by promoting a state of ketosis and has been shown to facilitate weight loss without leading to overt decrement in athletic performance in most aerobic, endurance-related, or resistance activities. Though this can be utilized to an athlete’s benefit, especially those who must maintain strict weight classes, a weakness of this diet is that its restrictive nature negatively affects overall adherence rates, and the hypothesized sequelae of long-term adherence to KD, including dysregulation of lipid profiling/higher cardiovascular risk factors, physiologic manifestations including bone demineralization, increased cellular inflammatory responses, and decreased iron metabolism, can potentially increase mortality down the line. In addition, it is important to note that data about the potential interference of ketosis on signaling and regulation adaptation responses to exercise are currently lacking and these areas should be a future direction of research. Further research needs to be conducted on sport specific benefits and decrements as well as the ability for athletes to adhere to a strict version of a high-fat, low-carbohydrate, low-protein diet.

### 3.4. Non-Ketogenic Low-Carbohydrate Diets 

Given these stringent requirements, some athletes prefer to adopt a non-ketogenic low-carbohydrate diet to optimize function and performance. Low-carbohydrate diets are technically defined as carbohydrates composing <26% of total caloric intake—they include KD but allow for more liberal carbohydrate and protein consumption [59]. Low-carbohydrate diets such as the Atkins diet have been popularized in the media for weight loss, entering the realm of dietary fads in the United States as early as the 1960s [60].

Similar to KD, the goal of a non-ketogenic low-carbohydrate diet is to deplete glycogen stores, optimize gluconeogenesis, and increase contributions from fat oxidation to provide a larger source of energy. From a biochemical perspective, physical exertion during states of low glycogen increase phosphorylation of AMP kinase (AMPK), which phosphorylates PGC-1 α, thereby upregulating mitochondrial biogenesis [61]. However, in practice, no differences are observed in isometric strength, power, and strength endurance in trained gymnasts and taekwondo athletes while adhering to a low-carbohydrate, non-ketogenic diet. These athletes were sustaining moderate to vigorous intensity (50–84% of one-repetition maximum) on 3–4 weeks of a this diet (<25 g carbohydrates, 55% fat, 41% protein) compared to a regular Western diet (47% carbohydrates, 39% fat, 15% protein) [58,59]. 

An important note is that while experimental group diets in these studies are labeled as “ketogenic”, when observing the macronutrient distributions, these diets are more reflective of a low-carbohydrate diet. While even peer reviewed research studies are titling diets as “ketogenic”, a true ketogenic diet is restrictive of carbohydrates and protein with a fat content of at least 75% of the total diet. Studies that label diets as ketogenic with more liberal protein and carbohydrate distributions than true ketogenic diets highlight the difficulty in terminology of research regarding ketogenic vs. low-carbohydrate diets. In a study that assessed body composition and athletic performance outcomes in ten Taekwondo athletes randomized to a reported KD group versus ten Taekwondo athletes randomized to a non-KD group, those adhering to carbohydrate restriction had noted improvement in a 2000-m timed run and increased weight loss, in addition to subjective reports of feeling less fatigued [62]. Though this paper reports the outcome of a presumed KD, when looking into details of the diet of the KD group in particular, the diet was actually limited to only 55.0% fat and had a high 40.7% protein intake, which would technically make this diet low-carbohydrate instead of KD. Regardless, this improvement in sprint times and perceived exhaustion was notable. Lastly, one study showed that strength-trained males who adhered to a low-carb, high-fat diet (defined as no more than 40% carbohydrates per day, protein level set at 2 g/1 kg of fat free mass, calorie level set at 35 kcal/kg) for 12 weeks had higher BMI and free fat mass than the low fat, high carbohydrate group, which has been a similar conclusion drawn from athletes adhering to KD [63].

Though adherence rates may be improved with the low-carbohydrate diet, there is a paucity of literature evaluating the role of a non-ketogenic, low-carbohydrate diet in athletes. One crossover-controlled trial enrolling 24 participants (6 of whom were controls) showed significantly lower peak performance and TTE when comparing the low carbohydrate group to the high carbohydrate group, in addition to no difference in peak VO2 max, heart rate, blood lactate, cholesterol (total, LDL, HDL) levels, production of pro-inflammatory cytokines, lean body mass, skeletal mass, and resting metabolic rate. Athletes in the low-carbohydrate arm of the study additionally had significantly increased caloric intake and higher blood ketone levels (beta-hydroxybutyrate), thus indicating a greater predisposition to be in ketosis and to optimize fat oxidation, but no comparison is made to athletes on a true KD [64].

To investigate the role of carbohydrate restriction on performance, another study divided 18 participants into a HI-HI group (consumed 195 g of carbohydrates) or a HI-LO group (17 g carbohydrates). The carbohydrate restriction occurred during the three-hour period in between sessions prior to the 250-kJ time trial. The low-carbohydrate (HI-LO) athlete group had significant improvement in 250 kJ time trial compared to the HI-HI group [65]. While performance improvements between groups varied, expression of mitochondrial enzymes, such as citrate synthase and cytochrome c oxidase, did not significantly differ between groups [62,66]. This was one of the first studies to support the theory of “train low, compete high”—training while glycogen levels are low, and competing when glycogen stores are repleted—promoting improvement in exercise capacity in athletes by physiologically optimizing the body for performance. Though there are proposed benefits from a low-carbohydrate diet, it is important to acknowledge uncertainty and pitfalls as well—a long term, observational study noted higher mortality rates and increased rate of cardiovascular risk factors in a low-carbohydrate diet with a higher-than-recommended intake of saturated fats as well as nutrient deficiencies [67]. 

Furthermore, an additional non-ketogenic, low-carbohydrate diet regimen is commonly known as the protein-pacing diet. The protein-pacing diet is defined by intake of 1.4 to 2.0 g per kilogram of body weight protein per day (about 20–25 g of protein per meal), for 5–6 meals per day. When combining this diet with resistance, interval, stretching, and endurance (RISE) training over a twelve-week period, notable improvements are seen in muscular strength/power, measured via sit-ups, push-ups, and bench throws in a group of thirty exercise-trained women [68]. Similar improvements were seen in exercise-trained men, namely improved core maximal strength (1-RM), power (squat jumps), aerobic performance (5 km timed trials), flexibility (sit-and-reach), and overall cardiovascular health (diastolic blood pressure and augmentation index), thus illustrating notable physical and cardiovascular benefits from protein-pacing with RISE. These benefits were hypothesized to be related to increased release of insulin-like growth factor (IGF-1) [69]. Since the protein-pacing diet has known benefits in improving body weight and total abdominal fat, this option could be a beneficial alternative for overweight or obese individuals who seek performative cardiorespiratory benefits, such as improvements in VO2 max [70], in addition to cardiometabolic benefits which promote ongoing physical activity and contribute to gains in upper and lower body strength, endurance, flexibility, and balance [71].

In conclusion, low-carbohydrate diets do not show any changes in athletic performance; however, they carry the additional benefit of improved chance of adherence given its fewer restrictions. Similar to that of ketogenic diets, further research should investigate more sport-specific alterations to performance.

### 3.5. Vegetarianism and Plant-Based Diets

Vegetarian diets exist on a spectrum from not eating any type of animal or animal byproducts (veganism) to not eating any animal flesh but consuming eggs and dairy (vegetarianism) to regularly following a vegan diet, but occasionally consuming dairy, meat, fowl, or fish [72,73]. In 2003, approximately 2.5% of adults in the United States and 4% of adults in Canada follow vegetarian diets, but since then, interest in these diets has increased, with many famous athletes turning to these diets to enhance athletic performance.

When considering the implications of a vegetarian and vegan diets, it is important to consider the nutrient differences between these diets and how to best mitigate the pitfalls and maximize benefits of the vegetarian lifestyle. Overall, a large cross-sectional study showed that compared with omnivores, vegetarians and vegans consumed less total energy, saturated fat, cholesterol, and protein; however, they typically consumed more fiber and iron [74,75]. Poorly planned vegan diets, in particular, predispose individuals to protein deficiencies as well as vitamins B12 and D and iron, zinc, calcium, and iodine [72,74,76,77]. These deficiencies can manifest as fatigue and poor injury healing, which would all drastically impact athletic performance, recovery, and overall bone health. A recent study showed that vegan athletes had poorer bone quality compared to omnivores and had decreased benefit in bone quality from resistance training than omnivores as well [78]. While this study has not been further studied in additional athlete specific populations, this reflects a similar trend in a systematic review studying the general female population [79]. Further studies on athletes and the male population need to be completed. As a result, veganism, if not carefully monitored, can lead to suboptimal athletic performance and adverse health outcomes.

Something to additionally consider, however, is that compared with animal sources, plant sources of iron have less bioavailability so despite higher iron intake, individuals following a vegetarian or vegan diet have higher rates of iron deficiency, which can lead to fatigue and decreased athletic performance especially in endurance athletics [75,80]. Additionally, protein consumption is often a primary concern for those adopting a vegan or vegetarian diet [11]. Many focus on quantity of protein consumption, with many recommending at least 1.2–1.7 g/kg body weight per day (1.2–1.4 g/kg/day recommended for endurance athletes and 1.6–1.7 g/kg/day for strength and power athletes); however, this is not taking protein quality into account [81,82,83,84]. Despite the fact that soy can be high in protein and appears in the bloodstream quickly, even with similar quantities, there is conflicting evidence on the quality of soy protein. According to some studies, soy protein does not stimulate muscle protein synthesis in the same way that whey and animal protein sources would [75,85,86], while other suggest that many forms of soy protein contain high-quality protein. This may be attributed to either a lower leucine content, a key branched chain amino acid that leads to muscle protein synthesis [87,88], or different forms of processing and post-processing [89]. For instance, soy protein concentrates have increased protein absorption compared to soy flour due to processing. Of note, soy protein isolate did not show significantly higher protein quality compared to soy protein concentrate. However, some processing can increase protein quality, such as boiling soybeans at 100 °C [89]. More alterations and processes that impact soy bean protein quality can be seen in van den Berg et al.’s quantitative review [89]. Overall, it is important to be aware that vegetarian sources of protein can have varying levels of protein quality compared to meat protein due to both their contents and their processing.

If the source of soy or other vegetable protein were to be of lower quality, the amount of protein density or protein per calorie is lower and would requires more calories to achieve the same levels of muscle protein synthesis [88]. For example, obtaining 2.7 g of leucine from potato sources requires 1.4 kg (~10 cups) of cooked potatoes (1350 calories), while for the same amount of leucine, only roughly 300 g (11 oz) of chicken (495 calories) needs to be consumed [88]. Of note, older adult athletes especially have increasing demands for leucine and should aim for 2.5 g of leucine per meal to initiate muscle protein synthesis, which can be difficult to do with an all vegetarian diet [88]. Additionally, plant-based protein sources, such as legumes, nuts, millet, soy beans, cereals, and beans, contain additional components such as phytates (which bind protein to keep it in the digestive tract and not absorbed) and trypsin (which helps breakdown protein for absorption) inhibitors [90,91,92]. However, by home cooking or processing beans, the interaction of trypsin and trypsin inhibitors decreases and through soaking, fermentation, and germination, the phytate content in foods decreases, making plant-based proteins higher in quality [90,93]. That being said, interventional studies investigating whey vs. soy protein derivatives have shown negligible differences in lean mass development [94,95]. As a result of these differences in protein quality and leucine content between animal and vegetarian sources, a recent narrative review recommends that vegan athletes consume protein at levels of 1.4–2.0 g/kg/day [88]. 

Despite the practical differences in these diets, the literature to date does not suggest that plant-based diets are inferior to omnivorous diets. One systematic review that implemented a total of eight studies that studied aerobic and aerobic performance, strength training, and power and one study that investigated immune markers among endurance athletes suggested that there were essentially no differences in performance between the vegetarians and omnivores [96]. A recent study demonstrated that both vegan and omnivores have similar rates of myofibrillar protein synthesis, which could explain the similar output in power [97]. A more recent systematic review that included more recent studies, however, did highlight differences between the groups in VO2max, submaximal endurance time to exhaustion, creatine muscle content, fat-free mass, and skeletal muscle mass [98]. Among endurance athletes, there were mixed results stating vegetarians had increased relative VO2max [99,100,101,102], and one study suggesting an increase in submaximal endurance time to exhaustion [101]. Only Lynch (2016) et al. and Król et al.’s studies suggested an increase in relative VO2max, and both concluded that this was most likely due to lower body mass among vegetarian groups [99,100]. These studies concluded that there were additionally no differences in power output as well. Regarding endurance athletes, other studies included in the Pohl et al. review demonstrated no differences in power, power per lean body mass, endurance performance, or trainability [98]. For strength and power athletes, the majority of evidence shows that there was no difference in strength between omnivores and those following plant-based diets, except for one study showing decreased upper body strength in vegans [101] and another showing increased strength with knee extension in a plant-based diet group [103]. Other studies showed no differences in 1-rep maximums and power output; however, meat eaters had higher increases in skeletal muscle mass, fat-free mass, and creatinine muscle stores [98]. This suggests that plant-based athletes may benefit from creatine supplementation more than their omnivore counterparts. Overall, among all athletes, the consensus is that there is not a vast difference among different diets in their impacts on performance. The concerns among plant-based athletes are that they are at higher risk for decreased protein quality and intake as well as micronutrient deficiencies compared to omnivore counterparts. Both risks can be avoided through careful selection of foods as well as dietary consultations by registered dieticians and nutritionists.

In conclusion, there are a multitude of benefits in adopting a vegan or vegetarian diet, including decreased environmental impact or potential ethical concerns, and the purpose of this review is to demonstrate the potential benefits and pitfalls of adopting this lifestyle. However, without careful considerations, athletes could develop micronutrient deficiencies. Both the likelihood of developing these micronutrient deficiencies in athletes and the effect size that these would have on athletic performance are unclear. Further research should investigate this matter. This review serves to underscore the path to maximize its benefits and demonstrate that most studies show that, especially if carefully thought through in conjunction with a nutritionist or registered dietician, a vegetarian, vegan, or plant-based diet do not negatively impact athletic performance.

### 3.6. Intermittent Fasting

Intermittent fasting (IF) involves restricting eating times to certain hours of the day or fasting for a full day or multiple days at a time. Intermittent fasting has gained popularity among athletes due to endorsement from celebrities, professional athletes, and popular media [104]. The inspiration behind this diet option stems from principles of biochemistry; 3–8 h into a fasting period, glycogenolysis is initiated to maintain glucose levels in the blood, and there is an increase in fat oxidation to supply muscles with ATP; the body uses these as fuels for substrate until around the 36-h fasting mark, when protein catabolism begins [10,104].

With these biochemical principles in mind, IF protocols for the athlete population can be designed to promote glycogenolysis and increase ATP for muscles, which may have performative benefits. IF protocols vary in the designated hours of the day or days of the week in which an athlete restricts caloric intake. Some common examples include time-restricted eating (TRE), or the 16/8 diet, where individuals restrict intake for 16 h and eat ad libitum for 8 h, alternate day fasting (ADF), which involves alternating days of eating ad libitum and either completely fasting on alternate days or eating ~25–30% of energy needs, and whole-day fasting, or the 5:2 diet, consisting of a repeated cycle of eating ad libitum followed by days of complete fasting (5 days ad libitum, 2 days fasting) [105]. 

Many studies assessing the efficacy of intermittent fasting have been conducted in athletes observing Ramadan—one review showed a non-significant decrease in mean sprint time in athletes who were fasting from sunrise to sunset for the religious holiday, with no restrictions in caloric intake during the non-fasting time [106,107], while more recent data in a crossover study showed that sprint times during the fasting period were significantly decreased, even while controlling for other external variables, such as sleep and daily training regimen [108]. When the fast extended to several days, there are clear decreases in athletic performance. After a 3-day consecutive fast from dawn to sunset for 21 active males, deficits were noted in sprint speed and vertical stiffness in repeated sprints (2 sets of 5 × 5-s maximal sprints with 25 s of recovery, and 3 min of recovery between sets on a treadmill), though favorable cholesterol profiling was observed [109]. This study effectively serves as a smaller-scale fast when compared to a month-long fasting for those observing Ramadan.

A limitation to these studies involves confounders in studying athletes during Ramadan, including variations in sleep and variable caloric intakes when subjects can eat. A large systematic review analyzing intermittent fasting outside of the context of Ramadan, and involving primarily overnight fasts ranging from 8–12 h, found minimal difference in performance outcomes for exercise sessions lasting <60 min or >60 min in a majority of studies [110]. In a study that had athletes skip lunch, reduced TTE was noted in high-intensity cyclers over 10 days, in addition to significantly decreased absolute peak power output on day 2 of testing, but interestingly, there was normalization of performance by day 4 and beyond [111]. In a crossover study looking at 2000 m rowing trials, those who omitted breakfast (abiding by a 16:8 regimen) had significantly slower rowing times during the evening and increased RPE than those who ate breakfast, thus highlighting the impact of meal timing relative to activity in impacting performance [112]. When looking at a 20:4 regimen and its impact on an 8-week resistance program no significant difference was noted in total body composition and the cross-sectional area of muscle [113]. Of note, it is important to maintain a resistance training regimen during TRE, as a robust narrative review conducted by Aragon and Schoenfeld demonstrated that without resistance training, many who use TRE as a dietary strategy lose lean muscle mass over time [114]. Studies have not assessed the specifics of the 5:2 diet in humans, but early mouse models showed an improvement in endurance [115].

The risks of IF may outweigh the benefits in the athlete population, because a limited timeframe for eating may increase risk of developing low energy availability and lead to related consequences to health and performance [116,117,118]. It is already difficult for most athletes to meet the required caloric intake for muscle preservation and training performance, and it becomes much more difficult when one’s eating window is reduced. When fasting, the body maintains its energy levels by switching its primary fuel source from carbohydrates to fats. This increases fatty oxidation and promotes ketosis, similar to the ketogenic diet. Thus, IF may be useful for weight loss and athletes who must maintain a strict weight class; however, it may lead to decrements in other aspects of performance, as seen in within-day energy deficits associated with metabolic compensation in elite collegiate swimmers and male endurance athletes, and menstrual dysfunction in female endurance athletes [116,117,118]. The varying results seen in the athlete population suggests more studies need to be conducted to elucidate the effect of intermittent fasting on performance. This is complicated that many of the studies conducted are on those participating in Ramadan, which can impact its generality. Furthermore, given the numerous variations of IF protocols, research should be conducted on the proper protocol for athletes.

### 3.7. Disordered Eating and Eating Disorders in the Trained Athlete

Increased pressures to lose weight or maintain a low body weight or lean physique for the purpose of enhancing athletic performance may increase the risk of following a restrictive diet, disordered eating, or a clinical eating disorder, which can hinder performance as well as be detrimental to physical and mental health. Eating disorders are clinical conditions characterized by abnormal behaviors and attitudes toward eating and food. Disordered eating is a descriptive phrase rather than diagnosis that describes eating behaviors that contain aspects of certain eating disorders; however, they do not fit all the clinical criteria to meet a specific eating disorder. Studies indicate that a higher proportion of athletes, compared to non-athletes, exhibit disordered eating and eating disorders, and certain athletes are more at risk than others [119,120,121]. One study suggested that 13.5% of athletes had subclinical or clinical eating disorders compared to 4.6% in a non-athlete population [122]. Disordered eating and eating disorders exhibit the highest frequency in sports that emphasize leanness and aesthetics, such as gymnastics and ballet, endurance sports, as well as sports with weight classes, such as wrestling [122]. Men and women have different rates of disordered eating and eating disorders. An overview of eating disorders seen in athletes include: Anorexia Nervosa: An eating disorder characterized by extreme food restriction and an intense fear of gaining weight, leading to significantly low body weight [123].Bulimia Nervosa: An eating disorder characterized by recurrent episodes of binge eating followed by unhealthy compensatory behaviors to prevent weight gain [123].Binge Eating Disorder: An eating disorder characterized by repeated episodes of binge eating, or excessive food intake [123].Orthorexia Nervosa: An eating disorder characterized by a fixation on healthy eating and fear of eating foods perceived as unhealthy [124].

The effects of disordered eating on athletic performance and overall health can be significant. Eating below one’s nutritional needs may lead to muscle weakness, fatigue, injury, and an unusually long injury recovery time [125], which, over time, may advance to anemia, amenorrhea, and early-onset osteoporosis. Overall, research suggests more restrictive diets and eating patterns may increase risk of developing disordered eating and eating disorder in athletes [126,127]. Therefore, diets that restricts one or more food groups, such as the ketogenic diet or plant-based diets, or restrict eating windows, such as intermittent fasting, may appeal to athletes with disordered eating attitudes or lead athletes toward disordered eating behaviors. Less restrictive diets, such as the MedDiet, may reduce disordered eating attitudes and behaviors; however, the research on this is inconclusive. A 2015 study found that higher adherence to the MedDiet decreased the risk of developing binge eating disorder in a non-athletic population [128]. However, a 2022 study among a professional athlete population found that an increased adherence to the MedDiet was associated with a higher risk for developing orthorexia nervosa [124]. 

Before adopting one of the diets or eating patterns previously discussed, athletes should evaluate their reasoning for changing their current diet and closely monitor their health to avoid any negative impact to performance or overall health. Additionally, it is important for athletes to work with a registered sports dietitian or nutritionist before implementing a new diet to ensure their individual nutritional needs are adequately met. Overall, an athlete’s diet must provide enough energy and nutrients for optimal health and performance and should avoid any restrictive eating patterns that promote disordered eating behaviors. 

## 4. Conclusions

The recent position statement from the American Dietetic Association, Dietitians of Canada, and American College of Sports Medicine stated that physical activity, athletic performance, and recovery from exercise are enhanced by optimal nutrition [1]. Athletes at all levels are becoming increasingly aware of the links between nutrition and diet and athletic performance thanks to increasing levels of research in the fields as well as the endorsements of professional athletes. The increase in interest and prolific research prompted a critical analysis of recommended diets for athletes to highlight the status of research on how different diets have been shown to affect performance and potential pitfalls of each diet. The role of this paper is not to take the place of consultation from a dietitian or endorse wholesale change in diet, but rather highlight research on how athletes may be able to analyze their own diets and see what alterations, if any, should be made to impact performance. 

In summation, the MedDiet has some of the most robust research on diet adherence and improvements in performance and has been shown to lower inflammation while providing adequate nutrition to potentially help athletes with recovery and continue training. Low-carbohydrate diets and ketogenic diets have different distributions of macronutrients, with low-carbohydrate diets having more liberal guidelines of protein and carbohydrates, although many conflate the two diets. Both have not been shown to be detrimental to athletic performance, with low carbohydrate’s easier restrictions possibly leading to higher adherence. Vegans and vegetarians are at increased risk of micronutrient deficiencies, such as vitamins B12 and D and zinc, iron, and calcium, and may have lower leucine content, which has been shown to be key to muscle protein synthesis. Plant-based athletes should err on the side of increased range of protein intake compared to omnivore athletes, but individuals on plant-based diets have not shown decreased performance. Lastly, intermittent fasting, although proven to promote weight loss, has been shown to decrease athletic performance in both endurance and aerobic sports performance. These recommendations are highlighted in Table 1, and mock diets of each of these diets are presented in Table 2, Table 3 and Table 4.

Whatever the diet an athlete subscribes to, we recommend that athletes consume 6 to 10 g·kg^−1^ body weight·d^−1^ (2.7–4.5 g·lb^−1^ body weight·d^−1^) of carbohydrates, 1.2 to 1.7 g·kg^−1^ body weight·d^−1^ (0.5–0.8 g·lb^−1^ body weight·d^−1^) of protein (1.2–1.4 g/kg/day for endurance athletes and 1.6–1.7 g/kg/day for strength and power athletes), and fat intake should range from 20 to 35% of total energy intake for optimal performance [1]. Ultimately, consultation with a registered dietitian and other medical training staff is the proper way to highlight room for improvement in diet and providing appropriate recommendations that promote sustainable, actionable changes that can improve athletic performance.

## Figures and Tables

**Table 1 nutrients-15-03511-t001:** Definition, benefits, considerations, summary, and recommendations for popular dietary trends.

Diet Name	Definition	Benefits	Considerations and Risks	Research on Athletic Performance	Summary and Recommendations
Mediterranean diet (MedDiet)	Emphasis on intact grains, lean protein, olive oil, nuts, vegetables, and berries.	Anti-inflammatory, well-balanced, varied energy sources, low micronutrient deficiency risk, decreases risk of chronic diseases.	Overall healthy dietary pattern high in antioxidants and polyphenolic compounds, with an emphasis on whole foods rather than processed foods.	Higher adherence correlates with healthy body mass, increases in squat jump, power, muscle endurance, CMJ tests, Crossfit performance, VO2max, and better body composition.	Adherence does not increase burnout, the literature backs improvements in performance, balances improvements in body composition with adequate nutrition for training. Researched adherence assessment tools.
Ketogenic diet (KD)	Very low in carbohydrate, moderate protein, and high in fat. Classic is a 3-4:1 ratio of fat to carb + protein. Variations include up to 10% carbs, 75–90% fats, and 10% protein and allowing a state of ketosis. The literature often calls low-carbohydrate diets ketogenic.	Anti-inflammatory, weight loss/decreased body mass, improved cholesterol profile, decreased epileptic seizures, improvements in cardiovascular health.	Poor rates of adherence, potential decrease in muscle mass given overall decreased body mass, differences in effect of diet based on gender.	Efficacy in sports involving maintenance of weight classes by not leading to a decrement in resistance athletes, improvement in 2000 m run, reduction in respiratory exchange ratio.	Definitions in research varies, very difficult to adhere to in reality. Generally, no decrement in physical performance despite decrease in overall body mass.
Low carbohydrate	Defined as <26% carbohydrate intake per day	Anti-inflammatory, weight loss, higher blood ketone levels for presumed increase in fat oxidation.	Definition of low-carbohydrate diets in research is extremely variable and often includes ketogenic diets in study population. Can include highly processed “low carb” options with unhealthy levels of saturated fat and sugar substitutes; while “training low” may have some support, carbohydrates are an integral fuel source and performance factor for endurance athletic events.	Improvements in time to exhaustion, improvement in high-intensity interval training (faster 250 kJ trial time).	Difficult to isolate from the pure ketogenic diet in the literature. Similar outcomes to the ketogenic diet, but with improved adherence, proposed slight benefit in high-intensity exercise.
Vegetarian/vegan Plant-based diet	Primary energy source is plant-based with varied restrictions on animal products. Vegan are more restrictive and do not eat any animal biproducts, such as milk or eggs, while vegetarians do not eat any animal proteins.	A plant-based diet decreases the risk of chronic diseases, cholesterol, and saturated fat, and is higher in iron and fiber.	Deficiencies in vitamins B12 and D and potential deficiencies in iron, zinc, calcium, and iodine. Lower leucine content and protein density than omnivores. Healthy plant-based diets have high levels of healthy plant chemicals (polyphenolic compounds) and protective chemicals (e.g., resveratrol, which is a natural pesticide), shown to have health properties.	Possible increase in VO2max, submaximal endurance, no statistically significant decreases in performance compared to omnivores.	Intermixing vegetables high in possible deficiencies. Processing and post-processing as well as cooking techniques can affect protein quality in vegetarian sources such as beans and soy. Protein: 1.2–1.4 g/kg/day for endurance athletes and 1.6–1.7 g/kg/day for strength and power athletes.
Intermittent fasting	Restricted periods of eating to certain hours of the day or days of the week. Time Restricted Eating (TRE): Fasting for 16 h each day, 8 h eating ad libitum. Alternative Day Fasting (ADF): Fasting for 24 h at a time every other day. Whole-Day Fasting (WDF): Scheduled days of ad libitum eating and fasting.	Weight loss, anti-inflammatory properties, improved cholesterol profile, decreased blood pressure, improves insulin resistance.	Wide variety in the literature based on interval of fasting (hours to days), may not be suitable for those with underlying comorbidities (e.g., diabetes mellitus). Many studies in athletes observing Ramadan, which poses other external confounding variables in data.	Decrease in sprint speed and vertical stiffness in repeated sprints, reduced time to exhaustion in high-intensity cyclers, decreased rowing times and increased subjective fatiguability in rowers.	Generally, a decrease in athletic performance is noted in endurance sports, specifically in sprint speed, vertical stiffness, and rowing times. If doing TRE and wanting to avoid or minimize loss of muscle mass, ensure adequate protein and calories with resistance training. Many studies in athletes observing Ramadan

**Table 2 nutrients-15-03511-t002:** Plant-based sample diet. Aim for a minimum of 2000 calories per day.

Breakfast:
Tofu scramble (6 oz) with vegetables (1 cup)
2 slices whole-grain toast
Fresh fruit (1 cup)
*Total: 500 calories, 27 g protein, 14 g fat, 60 g carbs, 13 g fiber*
Morning Snack:
Peanut Butter and Jelly Sandwich
*Total: 380 calories, 13 g protein, 15 g fat, 47 g carbs, 5 g fiber*
Lunch:
Grilled tempeh or tofu (4 oz) with mixed vegetables (1 cup)
Hummus (1/4 cup) and 6 whole-grain crackers
1 cup fresh fruit
*Total: 456 calories, 22 g protein, 23 g fat, 36 g carbs, 11 g fiber*
Afternoon Snack:
Carrot sticks (1 cup) and hummus (2 tbsp)
*Total: 105 calories, 3 g protein, 6 g fat, 11 g carbs, 4 g fiber*
Dinner:
Brown rice (1.5 cup)
Black bean soup (1 cup)
Green salad with mixed vegetables (2 cups) and a vinaigrette dressing (2 tbsp)
*Total: 745 calories, 24 g protein, 18 g fat, 126 g carbs, 19 g fiber*
Dessert:
Non-dairy yogurt (1 cup) with peanut butter (2 tbsp) and granola (1/4 cup)
*Total: 400 calories, 16 g protein, 22 g fat, 42 g carbs, 5 g fiber*
** *Total for the day: 2586 calories, 105 g protein, 98 g fat, 322 g carbs, 57 g fiber* **

**Table 3 nutrients-15-03511-t003:** Mediterranean Mock Diet. Aim for a minimum of 2000 calories per day.

Breakfast:
Omelet with 2 eggs, spinach (1/2 cup), chopped tomatoes (1/4 cup), and feta cheese (1/4 cup)
*Total: 420 calories, 21 g protein, 31 g fat, 18 g carbs, 4 g fiber*
Morning Snack:
Handful of nuts
1 Banana
*Total: 275 calories, 7 g protein, 14 g fat, 32 g carbs, 6 g fiber*
Lunch:
Greek salad (4 cups) with grilled chicken (3 oz) and quinoa (1/2 cup)
Olive oil and balsamic vinegar dressing (2 tbsp)
*Total: 450 calories, 30 g protein, 20 g fat, 37 g carbs, 6 g fiber*
Afternoon Snack:
Hummus (1/4 cup) with 1 whole grain pita bread and carrot sticks (1 cup)
*Total: 282 calories, 9 g protein, 8 g fat, 46 g carbs, 11 g fiber*
Dinner:
Grilled salmon (4 oz)
Grilled vegetables (2 cups)
Whole-grain pasta (1 cup)
Roasted Chickpeas (1/2 cup)
*Total: 756 calories, 45 g protein, 14 g fat, 106 g carbs, 27 g fiber*
Dessert:
Whole milk Greek yogurt (1 cup) with Fresh Berries (1/2 cup) and Honey (1 tbsp)
*Total: 236 calories, 24 g protein, 11 g fat, 36 g carbs, 3 g fiber*
** *Total for day: 2419 calories, 136 g protein, 98 g fat, 275 g carbs, 57 g fiber* **

**Table 4 nutrients-15-03511-t004:** Low-Carbohydrate Diet. Aim for a minimum of 2000 calories per day.

Breakfast:
2 eggs cooked in 1 tbsp coconut oil
3 slices bacon
1 avocado
*Total: 588 calories, 24 g protein, 48 g fat, 14 g carbs, 11 g fiber*
Lunch:
Grilled Chicken Thigh (6 oz)
Mixed greens salad (1 cup) with vinaigrette dressing (2 Tbsp)
*Total: 531 calories, 39 g protein, 36 g fat, 18 g carbs, 3 g fiber*
Afternoon Snack:
1 oz almonds
Total: 164 *calories, 6 g protein, 14 g fat, 6 g carbs, 3 g fiber*
Dinner:
Grilled sirloin steak (8 oz)
One cup of steamed asparagus with 1 Tbsp butter
One baked sweet potato with 1 Tbsp butter
*Total: 723 calories, 53 g protein, 47 g fat, 20 g carbs, 5 g fiber*
Dessert:
1 cup full-fat yogurt with 1/2 cup berries, 1/4 cup nuts, and 1/4 cup unsweetened coconut flakes
*Total: 452 calories, 17 g protein, 35 g fat, 22 g carbs, 9 g fiber*
** *Total for the day: 2458 calories, 139 g protein, 180 g fat, 80 g carbs, 31 g fiber* **

## Data Availability

No data were accrued in the process of this study, and therefore, data availability statement is not applicable.

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
