# Peer review of "Popular Dietary Trends’ Impact on Athletic Performance: A Critical Analysis Review"

_nutrients, 2023, doi:10.3390/nu15163511_

Round 1

Reviewer 1 Report

I read the manuscript entitled " Popular Dietary Trends’ Impact on Athletic Performance: A Critical Analysis Review. " with great interest. The research design is appropriate and the methods are adequately described. However, the manuscript needs revision before it can be accepted for publication.

-       Lines 60-62, The main aim is clear and reported in the introduction; delete it from "results and discussion" and start reporting the real results.

-       Lines 63-70 Add some references.

-       Lines 56–120: These seem to be general and well-established considerations that probably fit better in the introduction; try to merge this part into the introduction.

-       Line 151: Please avoid "anaerobic and aerobic measures" because it’s too vague.

-       Line 179: Talking about adherence to the Med diet, mention that, in general, Med diet adherence in athletes seems to be higher compared with the general population (doi.org/10.1007/s11332-022-00899-z).

-       Line 213-214: The ketogenic diet was used for the first time as a medical therapy by Dr. Wilder in 1921 as a dietary approach to epilepsy as an alternative to complete fasting (Wilder, R.M. (1921)). The effects of ketonemia on the course of epilepsy Mayo Clin. Proc.)

-       Line 207: Several kinds of ketogenic diets have been well described here (DOI: 10.1017/S0007114521002609); however, the maximum gr of carbs is better than the percentage.

-       Regarding Vegan diets, recently, vegan diets have been shown to support exercised myoprotein synthesis rates similarly to omnivorous diets (DOI: 10.1016/j.tjnut.2023.02.023).

-       The roles of figures A, B, and C are not clear. Are there examples of diets? In this case, someone has 70 grammes of fiber per day. It’s not optimal for an athlete.

-       In general, results and discussion are too vague; try to provide for each section's (med diet, keto diet, etc.) weakness and strength, and at this point, what is well known and what we still don’t know, in order to suggest future investigations.

-        

-       Overall, the article is interesting, but at this stage, a practical take-home message and/or suggestions for novel investigations are missing.

Reviewer 2 Report

The manuscript entitled “Popular Dietary Trends’ Impact on Athletic Performance: A Critical Analysis Review” gives an overview on different dietary patterns impact on athletic performance. The article describes and critically evaluates the current level of evidence regarding popular diets and their impact on sport performance using mainly literature published between 2015-2022. Based on the available literature authors concluded that no one diet is universally recommend and that it is recommended for athletes to be aware of their nutrient needs through medical professional counsel and consider sustainable changes to achieve performance and body habitus goals.

In the Materials and methods section authors could explain in more details the search strategy behind papers selection. Except the mentioned database and the time frame for the paper extraction please provide additional information on the search strategy. Such as, type of papers, keywords used for obtaining the relevant publications, was there any exclusion-inclusion criteria for the paper selection, etc. What is the actual nature of this review paper? Is it simply a narrative review or systematic, integrative, scoping review let’s say?

In the context of Mediterranean diet, in the paper by Geric et al (2022) there is a tailored 2000 kcal, 7-day dietary plan based on Mediterranean diet principles and its cost per day worth of mentioning.

In regard to vegetarian/vegan diets there are body of evidence that suggest that in addition to beneficial effects such diets can also negatively affect health-related biomarkers. In line with that one should be very careful if starting to completely avoid foods from animal origin.

Round 2

Reviewer 1 Report

Dear authors, The paper has been deeply improved. I have only some advice to augment the quality of the manuscript before publication.

  • The introduction, despite being rich in details, is too long. Try to synthesise and go to the aim of the review and why it can help the actual state of knowledge.
  • Lines 178–180: ref. 31 is wrong; use doi.org/10.1007/s11332-022-00899-z.
  • Lines 186–189 reinforce your statement with DOI: 10.1017/S0007114521003202 and DOI: 10.1097/NT.0000000000000342.
  • Please check all references because often they don’t fit with your phrases.
  • Regarding the ketogenic diet, you can underline that data about the potential interference of ketosis on signalling and regulating adaptation responses to exercise is currently lacking.
  • Finally, considering that the aim of this article is to investigate the role of different dietary trends on athletic performance, the chapter on disordered eating and eating disorders may be off topic.

Author Response

The introduction, despite being rich in details, is too long. Try to synthesise and go to the aim of the review and why it can help the actual state of knowledge.

Thank you for this feedback. The authorship agrees that this introduction is too long. The information that is contained, however we believe is important to the counseling of athletes on diets. This information was moved down to lines 63-118 in a General recommendations section of the paper.

  • Lines 178–180: ref. 31 is wrong; use doi.org/10.1007/s11332-022-00899-z.

Thank you for making us aware. The citations became unlinked and have been corrected. See line 182

  • Lines 186–189 reinforce your statement with DOI: 10.1017/S0007114521003202 and DOI: 10.1097/NT.0000000000000342.

These citations have been added, see lines 189-191. Thank you.

  • Please check all references because often they don’t fit with your phrases.

    Thank you for making us aware. The citations became unlinked and have been corrected.

  • Regarding the ketogenic diet, you can underline that data about the potential interference of ketosis on signalling and regulating adaptation responses to exercise is currently lacking.

Thank you for this feedback. This information has been added to lines 326-328.

  • Finally, considering that the aim of this article is to investigate the role of different dietary trends on athletic performance, the chapter on disordered eating and eating disorders may be off topic.

Orthorexia and disordered eating are, in a sense, overly rigid dieting of any diet, and this directly affects athletic performance. Given the aim of this article is to inform athletes about the pros and cons of the diets and given their prevalence, especially among athletes, some of the readership will likely have struggled with or know someone who has struggled with disordered eating, we feel it is responsible to caution the athletes of the potential mismanagement of the information or recommendations.